# Isolation of Yeast and LAB from Dry Coffee Pulp and Monitoring of Organic Acids in Inoculated Green Beans

**DOI:** 10.3390/foods12132622

**Published:** 2023-07-06

**Authors:** Na Zhao, Mito Kokawa, Rasool Khan Amini, Weixue Dong, Yutaka Kitamura

**Affiliations:** 1Graduate School of Science and Technology, University of Tsukuba, 1-1-1 Tennodai, Tsukuba 305-8572, Ibaraki, Japan; zhaonaco@126.com (N.Z.);; 2Faculty of Life and Environmental Sciences, University of Tsukuba, 1-1-1 Tennnodai, Tsukuba 305-8572, Ibaraki, Japan; kitamura.yutaka.fm@u.tsukuba.ac.jp; 3Saza Coffee Holdings Ltd., 8-18 Kyoeicho, Hitachinaka 312-0043, Ibaraki, Japan; rasool.ameen@gmail.com

**Keywords:** isolation, fermentation, yeasts, lactic acid bacteria, coffee pulp

## Abstract

Yeast and lactic acid bacteria (LAB) are known to play an important role in the fermentation process of coffee post-harvest. This study aimed to isolate and screen yeast and LAB to be applied in lab-scale refermentation of commercial green coffee beans and coffee pulp with the aim of modifying the composition of organic acids (OAs) in coffee beans. Yeast and LAB strains were isolated from green coffee beans and dry coffee pulp and identified, and their effect on OA concentration in the coffee beans was quantified. In addition, the effects of different fermentation conditions (additional carbon source, different inoculum dose, and different types of coffee pulp) were evaluated based on OA quantification. Nine yeast isolates of *Rhodotorula mucilaginosa* and *Wickerhamomyces anomalus* were identified, and 11 LAB isolates of the species *Enterococcus mundtii* were identified. Of the 7 OAs quantified, quinic acid was the most abundant. The inoculation of isolated yeasts and LAB led to higher concentrations of OAs, showing the potential to realize modification of the OA composition of green coffee beans by re-fermentation with coffee-originated isolates.

## 1. Introduction

Coffee is one of the most traded agricultural commodities worldwide, with the annual world coffee consumption estimated to be over 166 million 60 kg bags in 2020/2021 (Trade Statistics Tables. https://ico.org/trade_statistics.asp?section=Statistics, accessed on 26 April 2023). Coffee is grown and harvested in coffee belt countries, with Brazil (32%), Vietnam (18%), Indonesia (6%), Colombia (6%), and Ethiopia (5%) as the top five producers [1]. On the other hand, the consumption of coffee is concentrated in non-coffee belt countries such as the United States, Germany, and Japan. Japan relies solely on imports to obtain coffee beans, ranking in third place among all importing countries with over 7 million 60 kg bags in 2020/2021. Coffee prices can vary greatly depending on the origin and variety, resulting in high expenses for importing countries such as Japan when purchasing coffee beans of high quality.

The quality of coffee beans is not only determined by the bean itself but also by the post-harvest processing it undergoes. Coffee beans are processed in local environments by wet-, dry-, and semi-wet process methods, during which fermentation occurs. Research on the fermentation process using fresh coffee cherries and fresh or processed coffee has been reported in most coffee-growing regions, such as Australia [2], Brazil [3], and India [4]. With the contribution of metabolic compounds generated by microorganisms, higher levels of flavor and aroma have been detected in the beans after fermentation [5,6,7,8,9,10,11]. Furthermore, the development of fermentation technologies has led to well-controlled coffee production methods, which have become valued as economical and reliable ways to obtain coffee beans with higher quality. However, there is a lack of research regarding coffee fermentation in non-coffee belt countries such as Japan.

One of the ways in which the fermentation process can be controlled is to use starters. The use of starter cultures during coffee processing has been shown to directly affect the sensory profiles of roasted beans [5]. Recent studies on coffee fermentation have mainly used yeast and lactic acid bacteria (LAB) as starters for fermentation. Some of the yeasts most studied in coffee fermentation include *Saccharomyces cerevisiae*, *Pichia kluyveri*, *Hanseniaspora uvarum*, *Debaryomyces hansenii*, and *Torulaspora delbrueckii* [12,13,14]. In addition, *Wickerhamomyces anomalus* (formally: *Pichia anomala*) has been found to produce volatile compounds during fermentation [15] as well as contribute to the inhibition of undesired fungi.

LAB are also generally isolated from coffee farms [16,17,18,19] and support the fermentation process by producing antifungal compounds [20] as well as lactic acid. The lactic acid provides an acidic environment that is suitable for yeast to grow. At the same time, the lactic acid itself contributes to sensory characteristics in the coffee beverage, such as favorable acidity and a smooth taste. Different LAB species have been associated with coffee fermentation, such as *Leuconostoc mesenteroides* and *Lactobacillus plantarum* [16,20]. Even though diverse species and strains have been found, common characteristics such as acid and aroma production [11,19] and the potential ability to inhibit bacteria [20] and hydrolyzation during coffee fermentation [11] are similar.

Most of these starters have been isolated from fermentation processes which take place immediately after harvest, using the fresh coffee cherry [21,22]. During fermentation, mucilage is the key substrate for microorganism activities, which contains sugars for microbial degradation and modifies the flavor composition of coffee beans [5,23,24]. For coffee-consuming countries that do not grow the coffee crop, the fresh cherry is hard to access; thus, a different fermentation method that involves different fermentation substrates, as well as different strains, may be needed. Therefore, coffee pulp, the dried flesh of the coffee cherry, and a by-product during coffee production was used as a substrate in this study. The coffee pulp accounts for 29% of the dry weight of coffee cherries and is rich in sugars, caffeine, chlorogenic acid, trigonelline, diterpenes, and fiber [23]. Among these substances, sugars play an important role which is to be digested by microbial activities to generate organic acids (OAs) and aromatic compounds.

OAs are important chemical components in coffee beans and account for approximately 11% and 6% of green and roasted beans, respectively [25]. They have been investigated for their contribution to the quality and flavor of coffee beans, as well as an indicator of metabolic activity from microbial fermentation. A previous study found that higher concentrations of acetic and lactic acids after fermentation led to better sensory scores, such as sweetness, in fermented beans [26]. Not only do OAs contribute to the overall flavor of coffee beans, but they are also related to the formation of important aroma precursors such as amino acids and chlorogenic acids [27].

As a first step in the attempt to develop a novel pre-treatment process for green coffee beans, yeast and LAB were isolated from the fermented mixture using dry coffee pulp as the fermentation substrate, identified, and characterized as new starters based on the analysis of OAs. In addition, the effects of different fermentation conditions were evaluated based on OA production.

## 2. Materials and Methods

### 2.1. Coffee Beans and Pulp

Coffee (family *Rubiaceae*, genus *Coffea*) beans from Finca Los Tres Edgaritos-Cauca, Colombia (*Coffea arabica*, Castillo variety) and Nyeri/Karindundu, Kenya (*Coffea arabica*, SL N.28 variety) were used as samples for yeast and LAB isolation. The former bean variety was used for screening and evaluating the isolates. The coffee beans were processed by the wet method and dried to around 10% moisture content. Ripe and over-ripe pulp of Geisha variety were collected from same farm in Colombia and dried. The processed coffee green beans and pulp were shipped to Japan for the following experiments.

### 2.2. Spontaneous Fermentation of Coffee Beans with Pulp and Isolation of Yeast and LAB

The two varieties of coffee beans and dry ripe pulp were used for spontaneous fermentation. Spontaneous fermentation took place in 250 mL sterile blue-cap bottles with 5 g of green beans, 5 g of dry ripe pulp, and 50 mL of distilled water at 28 °C from 24 to 48 h. Two bottles were prepared as repeats for each type of bean.

Samples from the end of spontaneous fermentation were massaged for 3 min, mixed by vortexing (Lab companion, Jeio Tech, Seoul, Republic of Korea) for 2 min, then centrifuged at 1000 rpm for 5 min to separate large particles from liquid. After serial dilution by sterile distilled water, 100 µL of the liquid was spread on MRS agar (Biokar diagnostics, Solabia Group, Pantin, France) with 0.1% cycloheximide and 1% Calcium carbonate (CaCO_3_, FUJIFILM Wako pure chemicals corp., Osaka, Japan) for isolation of LAB and on YEPD (1% yeast extract (Biokar diagnostics, Solabia Group, Pantin, France), 2% peptone (Kyokuto, Japan), 2% glucose (FUJIFILM Wako pure chemicals corp., Osaka, Japan), 2% agar (FUJIFILM Wako pure chemicals corp., Osaka, Japan), adjusted pH to 3.5) for isolation of yeasts. Plating was performed in triplicate. The MRS agar plate was incubated under anaerobic conditions in square jars (AnaeroPack, Mitsubishi Gas Chemical Company, Inc., Tokyo, Japan) at 30 °C for 48 h. The YEPD plate was incubated in aerobic conditions at 28 °C for 48 h.

The colonies that appeared on the plates were purified by streaking 4 times or more on the same agar plates. The purified isolates were stored at −20 °C in MRS or YEPD broth containing 20% (*v/v*) glycerol for further identification.

### 2.3. Identification of Yeast and LAB Isolates

Purified LAB isolates were primarily identified by morphological characteristics (size, shape, color, height, and edge of colony) and their ability to produce acids, which was shown by the dissolution of CaCO_3_ around the colonies. In addition, only gram-staining positive and catalase-negative isolates were used for further nucleic acid identification [16]. Yeast isolates were directly applied to PCR for nucleic acid amplification after differentiation based on morphological characteristics.

For PCR analysis, reagents were prepared according to the protocol of AmpliTaq Gold™ 360 Master Mix (Thermo Fisher Scientific K.K., Tokyo, Japan). The 5.8S ITS rRNA and 16S rRNA gene regions of yeast and LAB isolates were amplified using the primers ITS1 and ITS4 [28] as well as 27f and 1495R [29], respectively. The amplified PCR products were separated by 1% agarose gel electrophoresis and purified with ExoSAP-IT™ PCR Product Cleanup (Applied Biosystems™, Thermo Fisher Scientific K.K., Tokyo, Japan) and outsourced for sequencing by Premix analysis (FASMAC, Kanagawa, Japan). The sequencing results obtained were aligned with sequences data in the GenBank by BLAST (https://blast.ncbi.nlm.nih.gov/Blast.cgi (accessed on 2 February 2022)).

### 2.4. Fermentation Process with Isolates

To screen LAB and yeast isolates for fermentation suitability, the isolates were inoculated to a mixture of green coffee beans, dry pulp, and water, and OA production during fermentation was monitored. Sterile distilled water (20 mL), green coffee beans (Colombia, 5 g), and dry ripe pulp (5 g) were mixed in sterilized 150 mL blue-cap bottles. The purified yeast and LAB isolates were separately re-cultured for 24 to 48 h in YEPD and MRS broth, respectively, to reach approximately 7 log colony forming units (cfu) mL^−1^ and resuspended in sterile distilled water. The isolates were inoculated to the mixture of beans, pulp, and water at a final concentration of 7 log cfu mL^−1^ and fermented for 72 h at 28 °C and 30 °C for yeast and LAB, respectively. The control group was prepared with the same procedure with the exception that no additional inoculation was added.

OAs in the coffee beans were sampled, extracted, and quantified as explained in the following section at 24, 48, and 72 h of fermentation for the yeast isolates and at 12, 24, 48, and 72 h of fermentation for the LAB isolates. Since OAs produced by LAB are known to be beneficial for their coexistence with yeasts [24], OAs produced earlier in the fermentation (12 h) were measured for the LAB isolates. The isolates were screened to choose those which gave high levels of individual OAs, as well as highest total OA content. For every fermentation condition, four replicates were measured.

The following 7 OAs: acetic, citric, formic, lactic, malic, quinic, and succinic acids were extracted and quantified with HPLC (LC-6AD Series, Shimadzu Corp., Kyoto, Japan) using the ion exclusion method by following previous research [30] with some adjustments. Extraction from the fermented beans was performed in a 50 mL Erlenmeyer flask with 1 g of green beans and 3 mL of 3 mM perchloric acid solution, and the mixture was vortexed for 10 min at room temperature. The mixture was then centrifuged at 14,000× *g*, 4 °C for 10 min. The supernatant was filtered using a 0.45 μm membrane filter. The filtered extract was stored at −20 °C until analysis.

The OAs were measured using the post-column method. The acids were separated with a SHODEX KC-811 column (8.0 mm I.D. × 300 mm) with SHODEX KC-G 8B guard column (8.0 mm I.D. × 35 mm) (Resonac Corp., Tokyo, Japan). A mobile phase of 3 mM perchloric acid was used at a flow rate of 0.6 mL min^−1^, and 0.7 mL min^−1^ of coloring reagent (ST3-R, Resonac Corp., Tokyo, Japan) was added post-column. The oven temperature was set at 50 °C and UV detector at 430 nm. Quadruplicate tests were run on all samples. All the procedures and experimental conditions of this section are shown in Appendix A.

### 2.5. Evaluation of OA Producing Ability of Selected Yeast Isolates under Different Growth Factors

After screening yeasts by high OA production, the following studies were carried out to optimize fermentation conditions. (1) Sucrose, as a general carbon source for yeasts as well as an important simple sugar in the coffee cherry, was added as an extra energy source to the fermentation bottle at a concentration of 1% (*w/w*). (2) To evaluate the effect of different inoculation volumes, higher population of 8 log cfu/mL was added at the start of fermentation. (3) Over-ripe pulp was used instead of ripe pulp as an alternative fermentation substrate. Each condition was evaluated with several selected isolates, and all other conditions were kept constant. The OA content in the beans was measured, as explained earlier.

### 2.6. Statistical Analysis

Analysis of variance (ANOVA) and post-hoc Tukey HSD analysis were performed with SPSS version 28.0.1.1 (IBM Corp. New York, NY, USA).

## 3. Results and Discussion

### 3.1. Isolation and Identification of Yeasts and LAB

A total of 70 isolates (53 yeasts and 17 LAB) were isolated from the coffee beans and pulp. These were identified to species level by sequencing analysis, and a total of 9 yeasts and 11 LAB were obtained. Strain information, nucleotide similarity, and reference accession numbers of the isolates are shown in Table 1. All isolates revealed more than 95% similarity in the 5.8 S rRNA and 16 S genes with published yeast and LAB strain sequences, respectively. Among the isolated yeasts, four isolates were identified as *Rhodotorula mucilaginosa* and five isolates were identified as *Wickerhamomyces anomalus*. All strains from *R. mucilaginosa* showed colonies that were orange or red in color, smooth, and highly moist, while strains from *W. anomalus* showed colonies that were of milk-white color, smaller in size, and dry. These yeast species are reported to be fermentative and have been found in soil, fruits, vegetables, and coffee [31]. In addition to their contribution to coffee fermentation, they have been shown to play a significant role in wine fermentation. For example, the co-fermentation of *R. mucilaginosa* with *Saccharomyces cerevisiae* significantly increased the content of the varietal aroma compounds as well as acids [32]. *W. anomalus* has also been shown to produce higher levels of acetates and ethyl esters, which supply a fruity note in wine fermentation [33]. All LAB isolates were identified as *Enterococcus mundtii,* which has been reported for potential usage in milk fermentation [34].

Studies have shown that strains that are close in terms of evolutionary distance may still show differences in fermentation behavior [28,35]. Thus, it is relevant to screen each strain based on its OA-producing ability. Therefore, the identified yeast and LAB were screened for their fermentation behavior in the following experiments.

### 3.2. Quantification of OAs during Fermentation with Inoculated Yeast Isolates

The nine yeast isolates identified in the previous section were screened to compare their ability to produce OAs. In the following results, the strains that were identified as *R. mucilaginosa* have been labeled with the suffix “R” (Y1R, Y5R, Y12aR, and Y12bR), while those that were identified as *W. anomalus* have been labeled with the suffix “W” (Y10W, Y18W, Y19W, Y38W, and Y43W).

OAs are known to play important roles in the sensory experience of coffee. It is widely recognized that acidity and the resulting perceived sourness are key to coffee quality [25]. There are naturally presenting OAs in coffee fruits, as well as those maintained after fermentation, such as citric, malic, quinic, and succinic acids. Being regarded as the main acids in green coffee beans, these OAs favor the sensorial characteristics of coffee [5,30]. The seven targeted OAs measured: acetic, citric, formic, lactic, malic, quinic, and succinic acids, are considered to display pleasant and favorable flavors such as flowery, fruity, and acid flavors.

All seven OAs were detected in all beans inoculated with yeast isolates throughout the fermentation period (Figure 1a, Appendix A). Quinic acid was the most predominant acid in the fermented green beans, while succinic acid showed the lowest content among the acids measured.

Although there was no clear trend in the increase or decrease of each OA, the highest values were found in the later fermentation stages. Acetic acid, which gives an acidic and clean flavor to the coffee, was found to be most abundant in the Y12bR incubated beans after 72 h fermentation (1.33 ± 0.28 mg g^−1^). Citric acid is generally considered to show citrus and acidic flavor and can assist with flavor modification as well as increasing cupping scores by spontaneous fermentation [36]. The highest citric acid (0.82 ± 0.22 mg g^−1^) was detected in beans inoculated with Y12aR at 48 h. Formic acid, which gives a sour and bitter taste as well as adds a fermented aroma, was highest (0.49 ± 0.05 mg g^−1^) in the control group (no additional inoculation) at 24 h. Lactic acid has a sour, acidic taste and is known to contribute to a “full body” mouthfeel and “smooth” sensations [19]. The highest lactic acid was detected in Y1R fermented beans at 72 h (2.41 ± 0.03 mg g^−1^), and the second highest concentration was detected in Y12bR inoculated beans at 72 h (2.21 ± 0.78 mg g^−1^). Malic acid gives out a sour taste and, together with lactic, citric, acetic, and fumaric acids, may benefit in increasing cupping score as well as increasing fruity flavor [19]. Among all fermented beans with yeast isolates, only the inoculation of Y1R, Y12aR, and Y12bR showed a significant difference with control, with Y12aR inoculated beans showing the highest content of malic acid (0.57 ± 0.07 mg g^−1^) at 48 h. Quinic acid was the most abundant among all OAs and was detected to be highest (2.36 ± 0.65 mg g^−1^) with Y12bR inoculation at 48 h. Quinic acid not only adds an acidic taste to the beverage but, along with succinic acid, also contributes to a perceptible bitter taste [25,37]. Succinic acid showed the lowest concentration among all measured OAs, with the highest detected amount in the control fermentation at 24 h.

Two-way ANOVA showed that the different *W. anomalus* and *R. mucilaginosa* strains significantly (*p* < 0.01) affected the amount of quinic acid during fermentation. The content of individual OAs did not show a clear increasing or decreasing trend during fermentation.

The total concentrations of the seven OAs measured are shown in the bottom row of Appendix A. After 24 h of fermentation, all inoculated groups showed higher total OA content compared to the control group (3.51 ± 0.16 mg g^−1^), among which inoculation with Y12bR showed the highest amount (4.70 ± 0.44 mg g^−1^). At 48 h and 72 h, the highest total OA content was detected in the beans inoculated with Y12bR, with concentrations of 4.90 ± 0.64 mg g^−1^ and 5.25 ± 0.85 mg g^−1^, respectively.

### 3.3. Quantification of OAs during Fermentation with Inoculated LAB Isolates

OAs produced with the inoculation of 11 LAB isolates are shown in Figure 1b, and Appendix A. Green beans inoculated with L4 showed a high concentration of total OAs throughout fermentation, with the highest content among other inoculations at 12 h and 72 h of fermentation, and the second highest content at 24 h and 48 h of fermentation. The production of OAs by LAB would be beneficial when considering their coexistence with yeasts. Researchers have found that acidification by LAB not only aids the growth of starters via offering a low pH environment but also competes for space and nutrients with non-fermentation microbial species, thereby decreasing the generation of off-flavors [38]. Acidification was found to be accelerated by LAB fermentation in coffee pulp, which significantly increased lactic and fumaric acids by conversion from sugars (glucose and fructose) existing in the coffee pulp [19].

### 3.4. Evaluation of Different Fermentation Conditions on OA Production

The effect of the following fermentation factors: additional carbon source (sucrose), higher inoculation population, and different fermentation substrates (over-ripe coffee pulp) were evaluated based on the amount of OAs produced during fermentation (Figure 2). Since metabolites from microbial activity have been demonstrated to affect the final OA composition during the fermentation of green coffee [5,30], fermentation conditions that would influence microbial activity were evaluated.

An approximated 60% reduction of total simple sugars (glucose, fructose, and sucrose) is reported to occur during fermentation due to microflora activity [12]. Therefore, an additional carbon source (1% sucrose) was added to the initial substrate to stimulate the fresh cherry sugar composition with the expectation that it may enhance fermentation. For this evaluation, the strains with the highest and lowest OA production were used, which were Y5R and Y12bR as well as Y19W and Y18W as representative strains for *R. mucilaginosa* and *W. anomalus*, respectively. Figure 2a compares fermentation with and without sugar and shows the total OAs in the fermented beans along with the individual OA composition. Generally, the additional sucrose resulted in lower total OA production for beans inoculated with *R. mucilaginosa* strains. In particular, quinic acid showed a significant decrease with the addition of sugar. Similar effects were observed for *W. anomalus* inoculated beans, with the exception of Y19W inoculated beans. The Y19W strain produced the lowest content of OAs in the initial screening step, and it is interesting to observe that strains from the same species may have different preferences for carbon sources. The addition of sucrose did not increase OA content in the control group either, which was fermented without the addition of inoculates. Various other fermentable sugars (e.g., fructose and glucose, which originally exist in coffee pulp) could be considered as alternative carbon sources for coffee fermentation [39].

To enhance metabolic production of the yeast isolates, a higher inoculation concentration was added to the initial substrate (Figure 2b). Isolates that showed the highest OA production, namely, Y12bR and Y18W, were used. Conventionally, an inoculum of 6 to 8 log cfu mL^−1^ [40,41,42] is used for fermentation starters. Higher inoculum (8 log cfu mL^−1^) was shown to increase certain acids while decreasing others. Generally, higher inoculation did not efficiently increase OAs production during whole fermentation, which could be explained by energy and space competition from a large population. A similar phenomenon has been reported previously; yeast plateaued at around 8.1 log cfu mL^−1^ during fermentation [41]. Additionally, since achieving a higher population requires more culture time and cost, an inoculum of 7 log cfu mL^−1^ was sufficient.

Finally, the use of over-ripe pulp was compared to ripe pulp as a substrate for OA production (Figure 2c). The same isolates used for evaluating the effect of sugar addition were used. For all yeast isolates, the use of over-ripe pulp significantly decreased total OA concentration, mainly due to the decrease of quinic and citric acids. The difference was generally larger for the *R. mucilaginosa* isolates than for *W. anomalus*.

Overall, these results showed that fermentation using ripe pulp without sucrose and the inoculation of 7 log cfu mL^−1^ was sufficient for effective fermentation.

In this study, the isolated strains were inoculated onto the coffee beans and pulp without prior sterilization, thereby simulating the fermentation process occurring in coffee farms [6]. The microbial community existing in the material was kept intact, leading to a complicated balance with the inoculated strain [12,24]. Population dynamics of the inoculated strains as well as other microbial groups (e.g., acetic acid bacteria) that are involved in coffee fermentation, would help understand the contributions from each microbial community [38].

## 4. Conclusions

This study investigated the potential of novel isolates from coffee material. The isolates were evaluated based on organic acid (OA) production by fermentation with dry coffee pulp, which may help modify the flavors to coffee beans. In addition, alternative fermentation conditions were explored to maximize OA production during fermentation.

Initially, a total of nine yeasts (*Wickerhamomyces anomalus* and *Rhodotorula mucilaginosa*) and eleven lactic acid bacteria (LAB, *Enterococcus mundtii*) were isolated from the fermented mixture of green coffee beans and coffee pulp. These isolates were inoculated into a mixture of green coffee beans, pulp, and water, and fermentation took place while retaining the original microbial flora present in the beans and pulp. Fermentation with the inoculation of isolates affected OA content, which may also result in different volatile profiles from green and roasted beans, which will be measured in the future.

Additionally, alternative fermentation conditions such as the addition of carbon source, different inoculum dose, and different pulp were evaluated. On the basis of the OAs produced, the following conditions were sufficient for effective fermentation: no additional sucrose, inoculation of 7 log cfu mL^−1^, and the use of ripe pulp as the substrate.

Although this study focused on the production of OAs during fermentation, other chemical compounds that positively affect coffee flavor need to be monitored. Further research aiming to develop efficient fermentation conditions and investigate the correlation between metabolic activities and key aromatic components is expected. Finally, chemical composition changes induced by fermentation need to be assessed in both green fermented beans and roasted coffee beans to evaluate how starters influence coffee quality.

## Figures and Tables

**Figure 1 foods-12-02622-f001:**
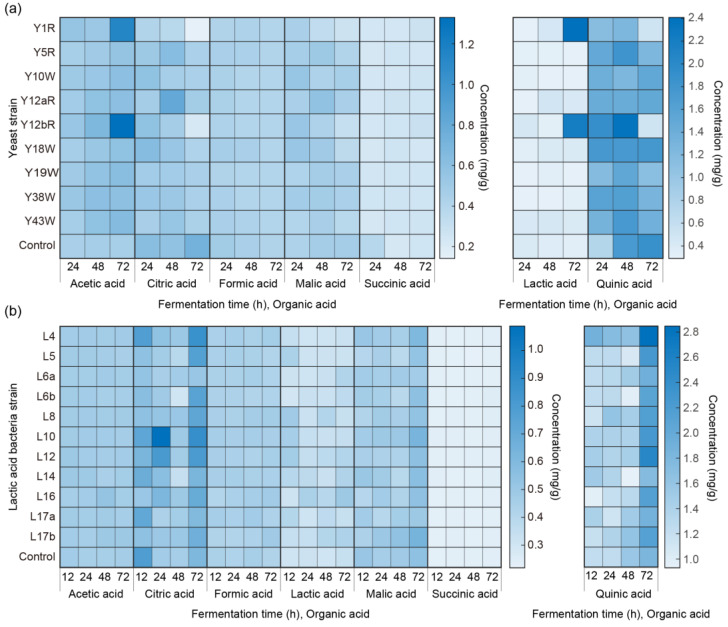
OAs quantified in fermented coffee with (**a**) yeast isolates at 24, 48, and 72 h and (**b**) LAB isolates at 12, 24, 48, and 72 h (*n* = 4).

**Figure 2 foods-12-02622-f002:**
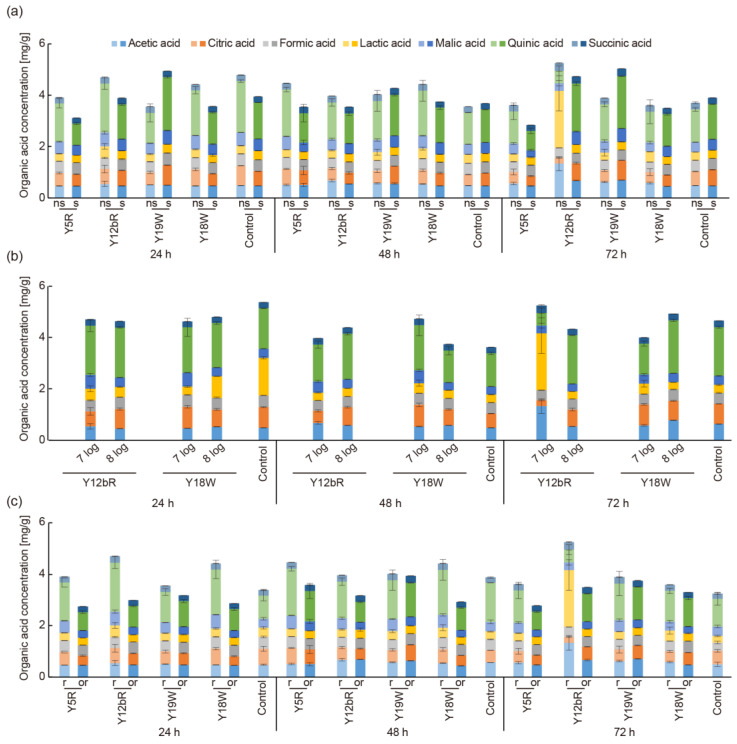
OAs quantified in fermented coffee at 24, 48, and 72 h with (**a**) additional carbon source, where samples with and without added sucrose are labeled ‘s’ and ‘ns’, respectively; (**b**) different inoculum doses of yeast isolates; and (**c**) fermentation with different types of pulp, where ‘r’ and ‘or’ stand for ripe and over-ripe pulp, respectively.

**Table 1 foods-12-02622-t001:** Alignment similarity of isolates with reference accession numbers and strain labels of the representative identified species.

Strain Label	Nucleotide Similarity (%)	Reference Accession Number	Species Information
Y1	97.80	OM523876	*Rhodotorula mucilaginosa*
Y5	99.83	MK646042	
Y12a	99.82	KF953903	
Y12b	98.94	ON242334	
Y10	100.00	MZ576855	*Wickerhamomyces anomalus*
Y18	100.00	MK343437	
Y19	100.00	MZ089535	
Y38	100.00	FJ713067	
Y43	100.00	MK757882	
L4	99.39	AP019810	*Enterococcus mundtii*
L5	100.00	AB831185	
L6	99.51	MZ869125	
L6b	99.75	MZ869177	
L8	98.53	MZ869152	
L10	99.50	KC985226	
L12	99.25	MZ869176	
L14	100.00	MN636722	
L16	99.75	CP029066	
L17	99.88	MW135231	
L17b	99.75	MW135237	

## Data Availability

Data is contained within the article or Appendix A.

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
