# Peer review of "Isolation of Yeast and LAB from Dry Coffee Pulp and Monitoring of Organic Acids in Inoculated Green Beans"

_foods, 2023, doi:10.3390/foods12132622_

Round 1
Reviewer 1 Report
The manuscript describes an interesting study directed to isolate, from dry coffee pulp, yeasts and lactic acid bacteria with the aim of modifying the composition of organic acids in Coffee Beans. The organization of the manuscript is adequate, English is good, some methodological details are missing and requires some clarifications, which are detailed:
- Genus and species of microorganisms, in italics. Check References
- p.2, line 47: countries
- p. 3, lines 104-5: replace "cultured" by "incubated"
- When a raw material is fermented, it is normally inoculated to have an initial concentration of 5-6 log orders UFC/ml or g. In this study, higher initial concentrations (7-8 log orders) are used, which is difficult for the starter to develop sufficiently to produce physicochemical modifications. This decision is not understood. On the other hand, the final viable cell concentration of the starter is not reported at the end of fermentation, as well as to see if the microorganism grow little or much. Please, clarify.
- 3.1: because of the 70 strains, were only selected 20 strains? Based on what things were selected?
-When you think about using new strains for food fermentations, it should then be proven that their use is positive for the sensory characteristics of the products. Isolation and study can be successful in the laboratory but can negatively influence products. Therefore, a sensory analysis of what is obtained after fermentation is necessary to know if isolated strains are really useful. Nothing of this aspect is reported in the manuscript
Reviewer 2 Report
Section 2.2: need to be more detail. Did the authors analyzed total indigenous bacteria on PCA?
Was there no washing process of coffee bean at all?
Line 102: Provide sufficient information (company, city,,,) on the culture medium YEPD agar.
Line 102: and on and YEPD : Delete ‘and’?
Line 111-113: Add the references
Line 127-134:
A clear description is needed regarding whether the cultivation of yeast and probiotics was conducted separately or simultaneously.
Furthermore, please express the inoculum concentration in percentage (%).
Is the inoculum concentration of 7 log based on an overnight culture?
Section 2.4: Please clearly present the procedure and experimental conditions of section 2.4 using figures or tables.
Line 190: Enterococcus
Enterococcus is currently being excluded from probiotics in some countries due to safety concerns. It is necessary to mention how Enterococcus strains are regulated in Japan.
Section 3.3, 3.4:
More comprehensive and in-depth discussions are needed in sections 3.3 and 3.4.
Please include a discussion on how the formation of acids affects the degradation of coffee's lipid components.
Additionally, expand the discussion on the contributions of each acid to the overall profile of coffee.
Please add the limitations of the current study.
Minor editing of English language required
Reviewer 3 Report
This study aimed to isolate and screen yeast and LAB (from green coffee beans and dry coffee pulp) for application in commercial green coffee beans and coffee pulp on a laboratory scale, with the aim of modifying the organic acid composition in coffee beans. The presented research results are extremely interesting and useful for practice. In my opinion, this manuscript is suitable for further processing by the editorial board of a scientific journal, as it has a high level of content and science. In my opinion, the text should be refined and missing information added.
Section " 1. Introduction" presents a current dataset on the importance of microflora for coffee bean quality, and on yeast and LAB isolates relevant to the fermentation process. In my opinion, more attention is needed to discuss the importance of lactic acid fermentation as well as other processes that occur during the spontaneous fermentation of green coffee beans. What types (species and genera) of lactic acid bacteria are involved in these processes? What are the biochemical properties of these bacteria? Which metabolites are most important in this case? What is the importance of organic acids?
Section "2. Materials and Methods" presents the analytical methods used. They are well described, but there is no data on the number of repetitions of each analysis. This is important for the statistical evaluation of the results. Was the study performed on only one batch of green coffee beans and dry coffee pulp?
Section "3. Results and Discussion" presents the results obtained. The modesty of the variety of yeast and LAB isolates is surprising. What could be the reason for this? Was it the isolation procedure used or the yeast and LAB culture parameters? Or was the number of batches of green coffee beans and dry coffee pulp too small? There is ample literature data demonstrating the presence and importance of many types and species of yeast and LAB in the spontaneous fermentation of coffee beans. In my opinion, these dates should be discussed in this manuscript.
Round 2
Reviewer 1 Report
The authors have responded satisfactorily to the observations made
Reviewer 2 Report
Accept. The manuscript would be suitable for publication in its current form.
minor editing for English language is required throughout the manuscript
Reviewer 3 Report
I wanted to reach out to express my sincere appreciation for the revisions made to the manuscript. These efforts have significantly improved the quality of the work, and I am pleased with the current version.
I am happy to inform that the manuscript is now ready to proceed to the next stage of the editorial process.